# RoBERTaEns: Deep Bidirectional Encoder Ensemble Model for Fact Verification

**Muchammad Naseer, Jauzak Hussaini Windiatmaja** , **Muhamad Asvial and Riri Fitri Sari** *

Department of Electrical Engineering, Faculty of Engineering, Universitas Indonesia, Depok 16424, Indonesia; muchammad.naseer91@ui.ac.id (M.N.); jauzak.hussaini@ui.ac.id (J.H.W.); asvial@eng.ui.ac.id (M.A.)
* Correspondence: riri@ui.ac.id

**Abstract:** The application of the bidirectional encoder model to detect fake news has been widely applied because of its ability to provide factual verification with good results. Good fact verification requires the most optimal model and has the best evaluation to make news readers trust the reliable and accurate verification results. In this study, we evaluated the application of a homogeneous ensemble (HE) on RoBERTa to improve the accuracy of a model. We improve the HE method using a bagging ensemble from three types of RoBERTa models. Then, each prediction is combined to build a new model called RoBERTaEns. The FEVER dataset is used to train and test our model. The experimental results showed that the proposed method, RoBERTaEns, obtained a higher accuracy value with an F1-Score of 84.2% compared to the other RoBERTa models. In addition, RoBERTaEns has a smaller margin of error compared to the other models. Thus, it proves that the application of the HE functions increases the accuracy of a model and produces better values in handling various types of fact input in each fold.

**Keywords:** fact verification; fake news; FEVER dataset; homogeneous ensemble; RoBERTa; RoBERTaEns

## 1. Introduction

Nowadays, fact verification models have been developed and implemented using machine learning [1–5]. These developments have improved the research aimed at detecting fake news in recent years [6–10]. Generally, a bidirectional encoder-based model has been developed to detect fake news to verify a fact. For example, the Bidirectional Encoder Representations from Transformers (BERT) [11] model has been successfully implemented to detect fake news [12,13] and fact verification [14,15]. Another model developed from BERT to detect fake news, called RoBERTa [16], has been successfully implemented [17,18]. RoBERTa has been implemented to verify a fact [19]. Some researchers found a new challenge in machine learning models to optimize and improve the performance of better methods, including applying ensemble models [20]. An ensemble is a classifier mechanism running concurrently that combines the predicted results of the components. Several cases of ensemble research carried out homogeneously have increased accuracy values in a model [21,22].

Ensemble deployments are becoming popular due to the advantages of using a single model across multiple datasets and machine learning tasks. In a simple case, by using an ensemble, a network is trained using a large dataset to check the result of the accuracy, loss, etc., in the validation set [23]. It is conducted to improve accuracy as a parameter to measure the success of the ensemble application for a model development [20]. Good fact verification requires the most optimal and has the best evaluation model. The goal is to make the news readers trust the reliable and accurate verification results.

One study [24] stated that RoBERTa has a higher accuracy value than the other models in verifying facts. Based on several previous studies, it has been proven that the homogeneous ensemble (HE) can improve model performance. This paper combined HE with

the RoBERTa model to claim a fact. We evaluated the application of the HE on RoBERTa to prove that the application of an ensemble can be used to improve the accuracy of a model. To obtain optimal results, we modified the algorithm structure of RoBERTa in the training and testing stages. Modifications made in the training phase can be used to set three samples on RoBERTa to improve model performance. Improved model performance using three samples has been proven [25]. Afterwards, the evaluation results of each model were summed.

Model accuracy and F1-Score were used to measure the performance of the method. In the experimental process, we combined three models of RoBERTa. We found no research related to fact verification by combining ensembles with the model, and we were interested in contributing to this topic to improve model performance in fact verification. Therefore, the results of the fact verification were expected to have better verification.

In summary, our contributions in this paper are as follows:

1. Improving a RoBERTa model by implementing the HE from three best parameters settings in RoBERTa, we call it RoBERTaEns.
2. Implementing one of the HE methods, i.e., bagging ensemble to RoBERTa, the kind of bagging ensemble used was called Sum.
3. Testing and comparing the performance of BERT, RoBERTa, XL-Net, and XLM models with RoBERTaEns to obtain the best model performance.

## 2. Related Works

The impact of fake news that is widespread in society becomes a challenge in the world of journalism when one must label a fact from the results of claims of whether news or information is true or false based on evidence. Nowadays, fact verification has become a very challenging task in the realm of Natural Language Processing (NLP) [26,27], and the research community has paid much attention to research issues related to fact verification [28–31]. There are three types of labelling a fact [32]: (1) If there is an evidence supporting a claim, the facts are labelled with SUPPORTS; (2) If there is an evidence refuting a claim, the facts are labelled with REFUTES; (3) If there is no evidence to support or refute the claim, the facts are labelled with NOT ENOUGH INFO.

There have been many studies related to fact verification for making a model and dataset [31,33] to test various types of models [34] for fact verification [35]. Many studies have tested various datasets to verify facts. For example, a study comparing two datasets [36], i.e., the Fake News Challenge (FNC) dataset and the FEVER dataset using the Recognizing Textual Entailment (RTE) model. One of the results of the research has been reviewed by using the FEVER dataset on the train and evaluation domains, and the accuracy of the FEVER dataset was better than the FNC. The FEVER dataset had an accuracy of 83.43% on the test results. Other studies have also implemented the FEVER dataset for fact verification [15,34]. The FEVER dataset has the subtasks of document retrieval, sentence retrieval, and claim verification [37].

In addition to comparing datasets, a comparison of models to obtain the best accuracy in verifying facts has been carried out by several researchers [14] who compared three models, such as (1) BERT [11]; (2) Enhanced LSTM (ESIM) [38]; and (3) the Neural Semantic Matching Network (NSMN) [39]. The research results prove that BERT has the best results, with an accuracy of 61%, for fact verification. Furthermore, a different study [16] developed the BERT model by changing the BERT training structure, resulting in a new model called RoBERTa.

The results of the research were conducted by comparing three models, such as (1) BERT; (2) Generalized Autoregressive Pretraining for Language Understanding (XL-Net) [40]; and (3) RoBERTa. This study shows that the result of RoBERTa has a higher accuracy than BERT, with an accuracy of 92.5%. These results attracted researchers to develop the RoBERTa model in order to produce novel or better classifications, one of which is a study implementing Natural Language Inference on the RoBERTa model [41]. The model architecture that has been developed is given in Figure 1.

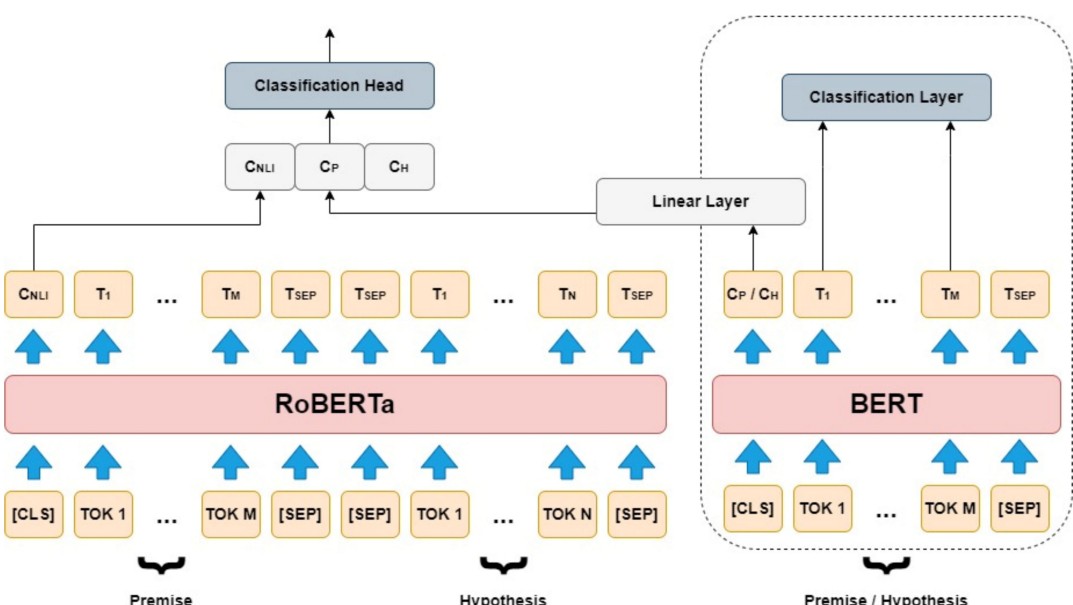

**Figure 1.** Architecture diagram of predicate-aware RoBERTa model.

The application of HE in several studies proves an improvement in model performance [25,42]. HE has a model architecture built with multiple base learners and is generated from a single learning algorithm, in contrast to a heterogeneous ensemble that is built with different algorithms. Many researchers in their research more commonly use HE compared to the heterogeneous one [43]. For example, recently, several studies have proven HE models to improve model performance. The study applied a tree-based HE model with feature selection to predict diabetic retinopathy. The results showed that HE was established to enhance the performance of the Support Vector Machine (SVM) model [42]. Another study [25] also developed a model by applying the HE to improve the evaluation performance of the Decision Trees, Logistics Model Tree, and Bagged Multi-layer Perceptron algorithms. As a result, an effective HE gave satisfactory results, with an average accuracy of 98.1%. The existence of various evidence of improving model performance by applying the HE makes us interested in examining the application of the HE on the RoBERTa model. The bagging ensemble method used in this study was Sum [44].

Many researchers have carried out the implementation of RoBERTa by combining datasets with the aim of verifying facts. A study concerning fact verification by combining the FEVER dataset and RoBERTa as a model for fact verification has been carried out [45]. This combination obtained an accuracy of 95.1% by using the F1-Score calculation. F1-Score is the average harmonic value obtained from the precision model. The majority of F1-Scores are used to evaluate the machine learning [46–48] model, especially for NLP [49]. With the application of the HE, the model applied for fact verification is expected to improve the performance of the RoBERTa model. One of the interesting results of the related works that we studied refers to the RoBERTa model and has an updated RoBERTa model explicitly by applying the HE. Our HE application is called RoBERTaEns. The application of HE in the transformer family has been applied [50,51] to detect hate speech. Meanwhile, the HE applied in our research focused on fact verification.

RoBERTaEns is formed from three RoBERTa-based models with different epoch settings and batch sizes. RoBERTa-m1 with epoch 5 and batch size 32, RoBERTa-m2 with epoch 30 and batch size 20, and RoBERTa-m3 with epoch 50 and batch size 32. The best epoch and batch size settings are from the research conducted in [52–54]. We chose RoBERTa because it has a better accuracy results based on several previous studies. We need the highest accuracy of the claim results to verify the facts to impact the news readers' trust in the news.

### 3. Methods

This section presents the proposed HE modification. As mentioned in Section 1, the main idea of applying the HE to RoBERTa is to improve the performance of the model. The more optimal the model, the more effective fact verification results for better fact verification. Algorithm modification in the RoBERTa model was applied at the training and prediction stages. At the training phase, we applied the HE to obtain the model predictions from the datasets. At the testing phase, we calculated the performance of the obtained model using accuracy, recall, precision, and F1-Score.

Furthermore, to calculate the significant value of the model, we used the 95% Confidence Interval (CI) [55]. CI indicates the probability that a parameter is below the mean value and measures the level of uncertainty or certainty in the sampling method. Cross-validation 10-fold times [56] were used for classification performance measurement in machine learning to obtain the CI values.

### 3.1. System Overview

RoBERTaEns reads input in sentences and encodes the sentences from the input. The sentences that have been encoded are matched with the data in the FEVER dataset. For example, "The chairwoman of Lockheed Martin and the President are currently the same person" which has the label "Supports" on the dataset. Each model with predefined parameters was combined with the HE using the bagging ensemble method. The bagging ensemble method that we used for classifying predictions was Sum. The bagging ensemble method is one of the HE methods that serves to create a better classifier by combining the respective predictions to form the final prediction. Figure 2 shows the process of applying the HE to RoBERTa to improve model performance.

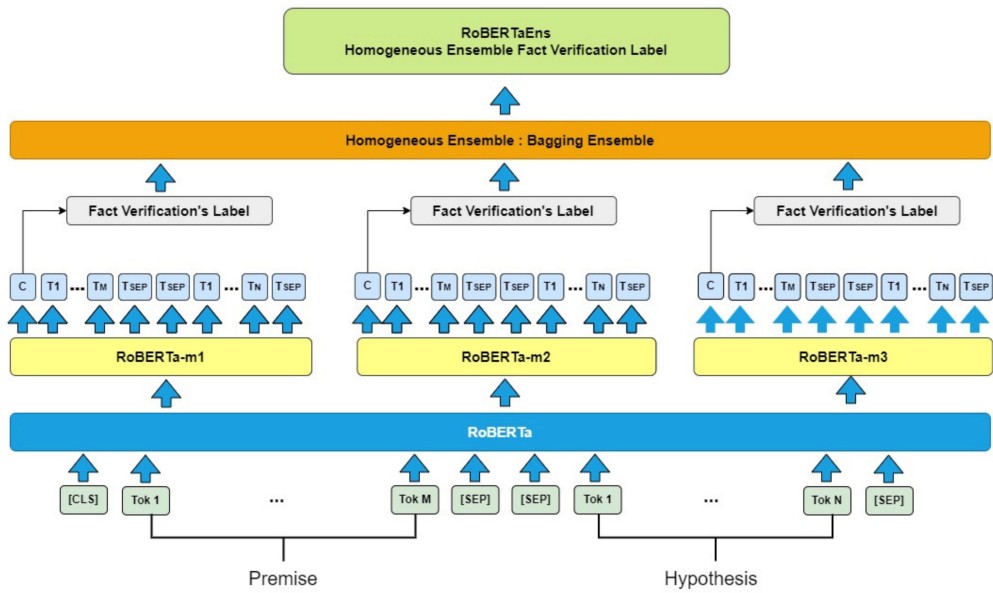

**Figure 2.** Block diagram of RoBERTaEns model.

Combined models created a higher learning rate for exploring parameters, and there is a potential for higher accuracy rates. A higher learning rate impacts the performance of the model in the sense that it takes less time due to the combination of the models [43]. It means that the application of the HE to the model potentially improves performance by requiring a shorter time than the time before HE was applied to the model. The results of the fact verification prediction are issued after the predicted value of the ensemble model is obtained. Each RoBERTa model was combined at the verification and prediction phases.

This section describes a method of improving our classification by applying the HE to RoBERTa. The first phase of the model read tokenized input sentences with a

[CLS] token at the beginning and a [SEP] token at the end of the sentences. [CLS] stands for classification and represents the sentence-level classification of news samples, [CLS] performed sentence representation during pretraining tokens used to decide sentence order. Furthermore, [SEP] separated sentences and made it easier for the model to know that the input consisted of several sample news sentences. The implementation of RoBERTaEns is conducted by combining the results of each RoBERTa with different epoch and batch size settings. The merging process was carried out using the bagging ensemble method. Each model produced model accuracy and labeled the verification of facts obtained from the classification. News classification was obtained from the output value of the neurons generated by the model. There are two output neurons generated: refutes and supports. The output neuron with the highest value indicates that the classification is in the highest neuron value class. If the resulting neuron output is the same value, it suggests that the news being tested is in another class, i.e., Not Enough Info (NEI). After obtaining the results for each model, the bagging ensemble works by combining the prediction results in each model and forming a new fact verification label. Figure 3 describes the flow diagram of the RoBERTaEns algorithm.

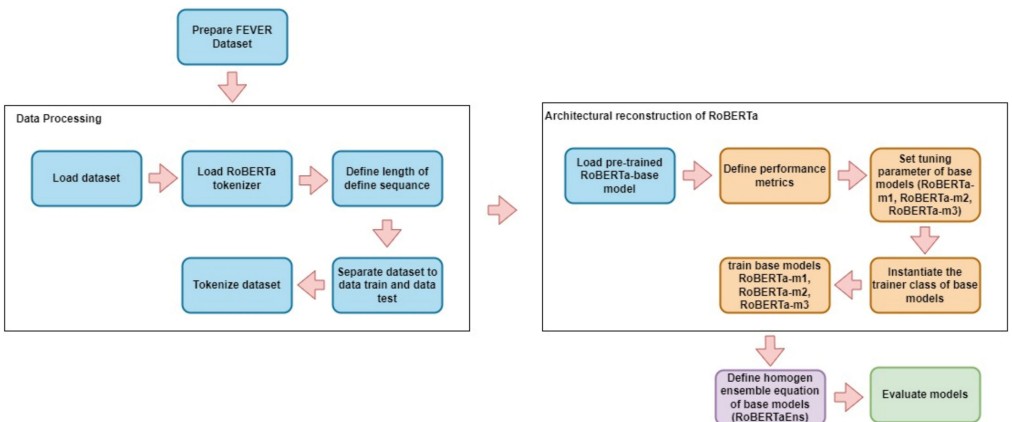

**Figure 3.** RoBERTaEns Flowchart.

*3.2. Training*

At this stage, the model architecture was combined, and the equation of RoBERTa was taken from a series of two segments (sequences of tokens), i.e., Question ($x$) and Answer ($y$) in BERT is $x_1, \ldots x_{Na}, y_1, \ldots y_{Ma}$. Usually, each segment consists of more than one natural sentence. Both segments are presented sequentially from a single input to BERT with tokens: $[CLS]$, $x_1, \ldots x_N$, $[SEP]$ and $y_1, \ldots y_M$, $[EOS]$. The RoBERTa model can be calculated as follows:

$$RoBERTa_{setup} = Ma + Na < Ta \tag{1}$$

where *Ta* is the parameter controlling the maximum sequence length during training, *Ma* is RoBERTa model, and *Na* is a sequence of *Ma*. *Ma* and *Na* are restricted. *Ma* is the RoBERTa model, and *Na* is the sequence of *Ma*. *Ma* reads the input token that is read by the RoBERTa model. Then, *Ma* is set to the epoch value and batch size in the model to produce a different number of tokens. *Ma* and *Na* are limited by *Ta*, so that the total *Ma* + *Na* is not more than *Ta*, where *Ta* is the parameter controlling the maximum sequence length of the original RoBERTa during the training process.

Furthermore, the input value of feature *X*, label *Y*, model (*Ma*), epoch (*E*) were tokenized using RobertaTokenizerFast. In the next step, we initialized and built the model. After the model was successfully created, it was run with the *X* and *Y* settings on the epochs that have been set. Algorithm 1 shows the training process in RoBERTa.

---

**Algorithm 1.** Pseudocode RoBERTa

---

1.    Input: Training feature *X*, training label *Y*, RoBERTa model *Ma*, epoch *E*
2.    Output: Trained RoBERTa models
3.    *X* ← tokenize *X* with RoBERTaTokenizerFast
4.    *Y* ← tokenize *Y* with RoBERTaTokenizerFast
5.    Initiate RoBERTa model *Ma* argument
6.    Construct RoBERTa model *Ma* architecture
7.    Compile RoBERTa model *Ma*
8.    Train RoBERTa model *Na* with *X* and *Y* on *E* epoch

---

The combination of the model was carried out on the training model by implementing the HE into the RoBERTa model. We called it RoBERTaEns. The regulatory architecture in RoBERTaEns can be calculated as follows:

$$RoBERTaEns_{setup} = Mensl + Nensl < Tensl \qquad (2)$$

where *Tensl* is the total parameter controlling the maximum sequence length during training, *Mensl* is the total model, and *Nensl* is the total sequence of *Mensl*. *Mensl* and *Nensl* are restricted. *Mensl* is a model of RoBERTaEns, and *Nensl* is an order of *Mensl*. *Mensl* reads the total input tokens from each RoBERTa model and sets the epoch and batch size values for each model to produce a different number of tokens. *Mensl* and *Nensl* are limited by *Tensl*, so the total of *Mensl* + *Nensl* is not more than *Tensl*, where *Tensl* is the parameter controlling the maximum sequence length of RoBERTaEns during training.

RoBERTaEns read the total of that tokenized using RobertaTokenizerFast. Afterwards, we initialized the model and run the model with the *X* and *Y* settings on *M*-repeated epochs on the RoBERTaEns with the *M* = 3. The results of each model in RoBERTa were summed for being processed in the HE, which aimed to improve model performance. Algorithm 2 shows the training process in RoBERTaEns.

---

**Algorithm 2.** Pseudocode RoBERTaEns

---

1.    Input: Training feature *X*, training label *Y*, RoBERTaEns models *Mensl*, epoch *E*
2.    Output: Trained RoBERTa models
3.    *X* ← tokenize *X* with RoBERTaEns TokenizerFast
4.    *Y* ← tokenize *Y* with RoBERTaEns TokenizerFast
5.    For i = 1 to M
6.    Initiate RoBERTaEns model *Mensl* argument
7.    Construct RoBERTaEns model *Mensl* architecture
8.    Compile RoBERTaEns model *Mensl*
9.    Train RoBERTaEns model *Nensl* with *X* and *Y* on *E* epoch
10.   End

---

*3.3. Parameters*

We tested two parameters for setting the model: epoch and batch size. Epoch carried out the learning process on NLP and batch size iterated on each training process. As previously discussed, in RoBERTaEns the three models were applied, i.e., RoBERTa-m1, RoBERTa-m2, and RoBERTa-m3. For each model, the different epochs and batch sizes were set to prove the best setting for fact verification and chose the best setting results of epochs and batch sizes for each model tested in previous studies [52–54]. The parameter settings of each model are summarized in Table 1.

**Table 1.** RoBERTa model parameter settings for ensemble.

| Model | Epoch | Batch Size |
|---|---|---|
| RoBERTa-m1 [52] | 5 | 32 |
| RoBERTa-m2 [53] | 30 | 20 |
| RoBERTa-m3 [54] | 50 | 32 |

*3.4. Prediction*

At this phase, the parameters in the model prediction were modified to adapt and implement the ensemble model. The predictions in the RoBERTa model were determined by setting one parameter that had been executed, while RoBERTaEns combined the prediction results from each run model. The sequence architecture of the RoBERTa and RoBERTaEns algorithms are shown in Algorithms 3 and 4, respectively.

---

**Algorithm 3.** RoBERTa predictions

1. Input: Test feature *X*, test label *Y*, trained RoBERTa model *Ma*
2. Output: Result
3. $X \leftarrow$ tokenize *X* with RobertaTokenizerFast
4. $Y \leftarrow$ tokenize *Y* with RobertaTokenizerFast
5. results < -predict *Na* with *X* and *Y*

---

**Algorithm 4.** RoBERTaEns predictions

1. Input: Test feature *X*, test label *Y*, trained RoBERTa models *Mensl*
2. Output: Final_result
3. Declare Array results
4. Declare Array final_result
5. $X \leftarrow$ tokenize *X* with RobertaTokenizerFast
6. $Y \leftarrow$ tokenize *Y* with RobertaTokenizerFast
7. For i = 1 to length (*Nensl*)
8.    results $\leftarrow$ append predict *Nensl* with *X* and *Y*
9. End
10. final_result $\leftarrow$ sum of results

---

The proposed model was evaluated using accuracy, precision, recall, and F1-Score. F1-Score performs a weighted average comparison between precision and recall. True Positive (TP) is positive news data detected correctly, True Negative (TN) is negative news data detected correctly, False Positive (FP) is positive news data classified as false by the system, and False Negative (FN) is negative news data classified as false by the system. The accuracy can be calculated as follows:

$$\text{Accuracy} = \frac{\text{TP} + \text{TN}}{(\text{TP} + \text{FP} + \text{TN} + \text{FN})} \tag{3}$$

Precision is used to identify positive cases with high false-positive values, which can be calculated as follows:

$$\text{Precision} = \frac{\text{TP}}{(\text{TP} + \text{FP})} \tag{4}$$

In contrast to precision, recall serves to identify positive cases with a high false-negative value that can be calculated as follows:

$$\text{Recall} = \frac{\text{TP}}{(\text{TP} + \text{FN})} \tag{5}$$

The F1-Score provides a harmonic average of precision and recall, which can be measured as follows:

$$F1 - Score = 2 \times \frac{(Precision \ \times \ Recall)}{(Precision + Recall)} \qquad (6)$$

*3.5. Dataset*

Our study used the FEVER dataset [31,57] because it yielded good accuracy for fact verification. In contrast to the other studies [40], which used the SQuaD, MNLI, SST, and RACE datasets to implement the RoBERTa model. The FEVER dataset is usually used for research based on fact verification [57,58]. The FEVER dataset has been proven to have the best performance and it is used in Claim Extraction and Fact Verification competitions [32,59]. The formation process for collecting the datasets has been carried out using a semi-automatic technique. Data labeling is done manually at the final stage of news labeling. The data labels of the claimed-results dataset are Supports, Refutes, and Not Enough Information. The data collected in the dataset are 10,000 data for the training phase and 10,000 data for the testing phase.

## 4. Result and Discussion

*4.1. Model Performance Results*

The results of RoBERTaEns proved that the application of the HE to RoBERTa improved model performance based on the results of accuracy and F1-Score that were obtained. RoBERTaEns combines RoBERTa-m1, RoBERTa-m2, and RoBERTa-m3 models to produce the HE. By testing several RoBERTa models and other models (BERT, XL-Net, XLM), the HE accuracy and F1-Score results are higher than the previously applied ensembles. In RoBERTa-m1, with an epoch setting of 5 and a batch size of 32, the best accuracy, precision, and F1-Score results were obtained by comparing RoBERTa-m2 and RoBERTa-m3. RoBERTa-m1 had an accuracy value 1.1% higher than RoBERTa-m2, a precision value 2.3% higher than RoBERTa-m2, and a F1-Score of 0.4%, which is higher than RoBERTa-m2 with an epoch setting of 30 and a batch size of 20. In addition, RoBERTa-m1 also produced the most of the measurements higher than RoBERTa-m3, with an accuracy value of 0.5% higher than RoBERTa-m3, a precision value of 1.2% higher than RoBERTa-m3, and F1-Score value of 0.2% higher than RoBERTa-m3. The results are summarized in Table 2.

**Table 2.** Performance of various RoBERTa model.

| Model | Epoch | Batch Size | Acc | Precision | Recall | F1-Score |
|---|---|---|---|---|---|---|
| RoBERTa-m1 | 5 | 32 | 77.3% | 82.3% | 84.0% | 83.1% |
| RoBERTa-m2 | 30 | 20 | 76.4% | 80.0% | 84.8% | 82.7% |
| RoBERTa-m3 | 50 | 32 | 76.8% | 81.1% | 84.9% | 82.9% |
| RoBERTaEns | | | 78.4% | 82.0% | 86.5% | 84.2% |

In this test, RoBERTa-m3 produced a recall value of 0.9% higher than RoBERTa-m1. RoBERTa-m3 resulted a recall value of 0.1% higher than RoBERTa-m2. Overall, RoBERTa-m1 was more dominant in obtaining accuracy, precision, and F1-Score. However, the highest recall was in the RoBERTa-m1 setting with 5 epochs and 32 batch sizes. Combining all models using the HE on RoBERTa improved the performance of the model for fact verification, i.e., it achieved an accuracy of 78.4%, a precision of 82%, a recall of 86.5%, and an F1-Score of 84.2%. The experimental results proved that combining several models can improve accuracy.

Figure 4 visualizes the comparison of the best RoBERTa model with RoBERTaEns (after the RoBERTa model was applied with HE). In our experiment, the performance of the RoBERTaEns model has higher accuracy and F1-Score values than other models. It proved that the implementation of the HE has succeeded in improving the performance of the model. However, the accuracy and F1-Score of the RoBERTaEns model is higher than RoBERTa-m1, RoBERTa-m2, and RoBERTa-m3.

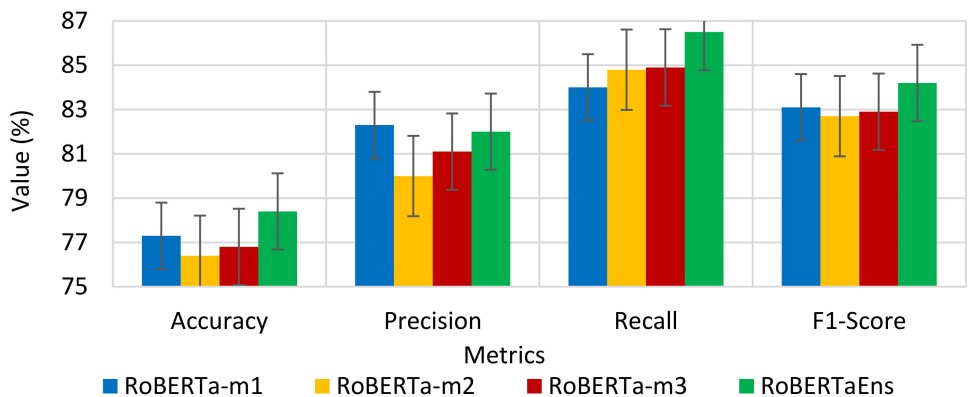

**Figure 4.** Performance of RoBERTaEns and various RoBERTa models.

In addition, we compared it with the other models, i.e., BERT, XL-Net, and XLM models. Each epoch and batch size setting in the model followed the setting with the best results in the comparison process of RoBERTa-m1, RoBERTa-m2, and RoBERTa-m3. The best epoch and batch size settings are 5 and 32, respectively. As a result, using the same dataset as RoBERTaEns, i.e., the FEVER dataset, RoBERTaEns has higher accuracy and better F1-Score than BERT, XL-Net, and XLM. The results of our comparison models are summarized in Table 3.

**Table 3.** RoBERTaEns model accuracy results compared to BERT, XL-NET, and XLM.

| Model | Accuracy | Precision | Recall | F1-Score |
|---|---|---|---|---|
| RoBERTaEns | 78.4% | 82.0% | 86.5% | 84.2% |
| BERT | 77.5% | 80.4% | 87.5% | 83.8% |
| XL-Net | 77.2% | 79.3% | 89.0% | 83.8% |
| XLM | 72.5% | 71.9% | 96.1% | 82.3% |

RoBERTaEns indicates that the correct positive prediction results are slightly lower than the comparison algorithm. However, the accuracy and F1-Score results are still higher than in BERT, XL-Net, and XLM. These results proved that RoBERTaEns has a measurement of the quantity that is closer to the actual value and its robustness is better than the three other models. Additional evidence of the HE's application on the model has been shown to improve the performance of the model we developed, i.e., RoBERTaEns. By applying bagging ensemble, RoBERTaEns improved its performance to produce a better classification. Figure 5 shows comparison results of the performance metrics.

*4.2. Fact Verification Result*

We have proven that RoBERTaEns produces a good model based on the model test performance and that it compares the results with the other models. We documented the results of the fact verification to find out whether it is claimed to be "Supports", "Refutes", or "NEI". The output neuron value generated by RoBERTaEns determined the label owned by the news to be claimed. RoBERTaEns required evidence sourced from datasets. Figure 6 shows an example in our experiment that required scattered evidence from multi-structured textual sources to conclude. Considering that the RoBERTaEns performance produces the best performance compared to the other models, the fact verification results generated by the RoBERTaEns model have a greater chance of producing a more precise fact verification.

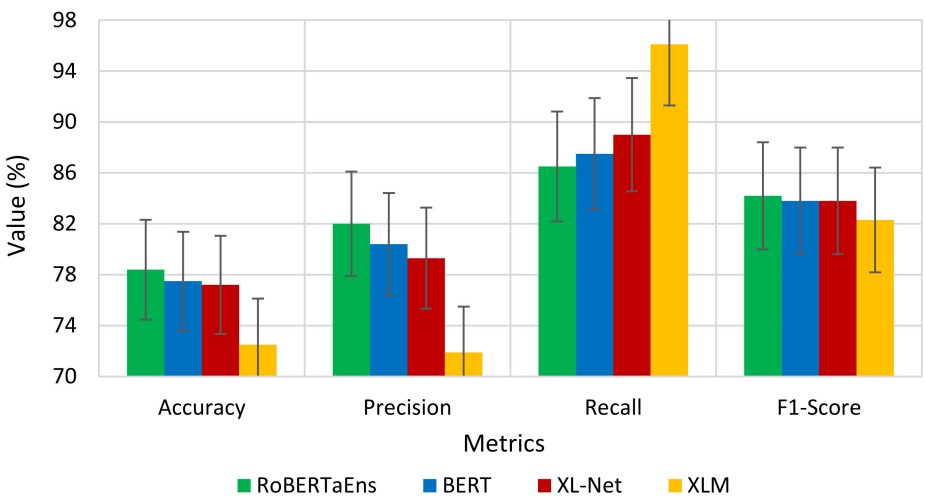

**Figure 5.** Performance of RoBERTaEns and other Transformer models.

| |
| --- |
| **Claim 1 :** Shooter is expert marksman who tries to stop the assassination of the president. |
| **Evidence Text Fragment :** Shooter is about an expert marksman who tries to stop the assassination of the president. |
| **Label : Supports** |
| **Claim 2 :** The Adventures of Pluto Nash is an Europe-American science fiction comedy game. |
| **Evidence Text Fragment :** The Adventures of Pluto Nash is an Australian-American science fiction action comedy game. |
| **Label : Refutes** |

**Figure 6.** Fact verification result.

We compared the performance of RoBERTa with RoBERTaEns after each datum in our dataset was grouped according to the following categories: Disease, Health, Political, Government, and Industry. The results of the performance comparison experiment are shown in Table 4. The F1-Score values for the "Government" and "Disease" categories in RoBERTaEns obtained the highest scores, with 98.90% and 92.27% respectively. However, the F1-Score value in the "Health" category had the lowest accuracy, indicating that this category is difficult to identify in the FEVER dataset. Various reasons for the low value obtained, among others, the amount of data in this category has a small sample which can cause errors of fact verification. Overall, we observed that RoBERTaEns has a higher accuracy and F1-Score values than RoBERTa-m1, except for the "Industry" category. In this category, the F1-Score obtained by RoBERTaEns has a lower accuracy of 2.28% and a lower accuracy of 1.75% compared to RoBERTa-m1. We used RoBERTa-m1 to compare RoBERTaEns because RoBERTa-m1 is better than RoBERTa-m2 and RoBERTa-m3. The complete results of accuracy (A), precision (P), recall (R), and F1-Score (F) for each topic category in the RoBERTa-m1 and RoBERTaEns models are shown in Table 4.

**Table 4.** Comparison of RoBERTa-m1 and RoBERTaEns by the News Topic Category.

| Topic Category | Parameters | RoBERTa-m1 | RoBERTaEns |
|---|---|---|---|
| Disease | A | 79.05% | 92.17% |
| | P | 78.48% | 93.27% |
| | R | 84.09% | 92.27% |
| | F | 81.19% | 92.77% |
| Health | A | 72.46% | 80.66% |
| | P | 74.26% | 81.29% |
| | R | 80.92% | 80.52% |
| | F | 79.51% | 80.92% |
| Political | A | 87.20% | 84.00% |
| | P | 88.98% | 82.60% |
| | R | 92.90% | 90.71% |
| | F | 90.26% | 88.90% |
| Government | A | 82.25% | 90.66% |
| | P | 88.90% | 91.86% |
| | R | 84.65% | 95.43% |
| | F | 86.71% | 98.90% |
| Industry | A | 74.35% | 72.60% |
| | P | 78.27% | 77.97% |
| | R | 80.54% | 78.53% |
| | F | 79.18% | 76.90% |

*4.3. Error Analysis*

The 95% confidence level is based on the CI shown in Table 5 with *n* (sample data) of 10,000 using 10-fold cross-validations. All the RoBERTaEns results show the lowest CI value compared to the other models. It validated our hypothesis that RoBERTaEns consistently outperformed the other models. These results also indicate that RoBERTaEns is more robust in handling various types of fact input in each fold.

**Table 5.** Margin of Error comparison using 95% CI.

| | RoBERTaEns | R-m1 | R-m2 | R-m3 | BERT | XL-Net | XLM |
|---|---|---|---|---|---|---|---|
| A | 0.01% | 0.02% | 0.01% | 0.01% | 0.02% | 0.06% | 0.10% |
| P | 0.00% | 0.04% | 0.01% | 0.02% | 0.05% | 0.12% | 0.02% |
| R | 0.01% | 0.10% | 0.01% | 0.01% | 0.14% | 0.04% | 0.09% |
| F | 0.00% | 0.02% | 0.01% | 0.00% | 0.05% | 0.13% | 0.05% |

**5. Conclusions**

Our research focuses on developing a new model based on three types of RoBERTa models for fact verification. We applied the Homogeneous Ensemble (HE) or a system based on a single classification approach for fact verification on the RoBERTa models with a FEVER dataset of 10,000 data. Our method is called RoBERTaEns. In this study, we found that the application of the HE on RoBERTa resulted in better accuracy and a higher F1-Score than RoBERTa-m1, RoBERTa-m2, RoBERTa-m3, BERT, XL-NET, and XLM. The experimental results showed that RoBERTaEns improves model accuracy by 1.1% compared to RoBERTa-m1, 2% compared to RoBERTa-m2, and 1.6% compared to RoBERTa-m3. The F1-Score results on RoBERTaEns also showed an increased percentage of 1.1% compared to RoBERTa-m1, 1.5% compared to RoBERTa-m2, and 1.3% compared to RoBERTa-m3. Regarding the stability of prediction accuracy, the ensemble application has a better fact verification than the other models. In addition, RoBERTaEns has a smaller margin of error compared to the other models, and it proves that RoBERTaEns produces more robust values in dealing with different types of fact input in each fold.

In future research, we suggest applying another ensemble method to the RoBERTa model, i.e., the heterogeneous ensemble, and comparing it with the HE. This exploration could be carried out on different datasets in future studies.

**Author Contributions:** Conceptualization, M.N., J.H.W., M.A. and R.F.S.; methodology, M.N., M.A. and R.F.S.; software, M.N. and J.H.W.; validation, M.N., J.H.W., M.A. and R.F.S.; formal analysis, M.N., J.H.W., M.A. and R.F.S.; investigation, M.N., M.A. and R.F.S.; resources, M.N., J.H.W., M.A. and R.F.S.; data curation, M.N., J.H.W., M.A. and R.F.S.; writing—original draft preparation, M.N.; writing—review and editing, M.N., M.A. and R.F.S.; visualization, M.N.; supervision, M.A. and R.F.S.; project administration, M.N. and R.F.S.; funding acquisition, R.F.S. All authors have read and agreed to the published version of the manuscript.

**Funding:** This research and the APC were funded by Universitas Indonesia under PUTI Doktor 2020, Grant number NKB-664/UN2.RST/HKP.05.00/2020.

**Institutional Review Board Statement:** Not applicable.

**Informed Consent Statement:** Not applicable.

**Data Availability Statement:** The dataset used in this study is publicly available at https://fever.ai, accessed on 1 November 2021.

**Acknowledgments:** We thank the Universitas Indonesia for financial support for this research. The authors would like to express their deep gratitude to the reviewers for their valuable suggestions and important comments that have greatly helped to improve the presentation of this manuscript.

**Conflicts of Interest:** The authors declare no conflict of interest.

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
