# Peer review of "RoBERTaEns: Deep Bidirectional Encoder Ensemble Model for Fact Verification"

_2504-2289, doi:10.3390/bdcc6020033_

Round 1

Reviewer 1 Report

This paper exploits the use of ensemble methods for the task of fact extraction and verification on top of a roberta based architecture.

The idea of this paper is relatively straighforward to get and the benefit of the ensemble method over the rest of the Roberta models, I would say is not significant given the size of the dataset.

Can you validate that?

Maybe if you can report confidence intervals.

The writing of the manuscript can be improved.

Some recent works on fact extraction and verification are missing. For instance see the review paper here 

Bekoulis, Giannis, Christina Papagiannopoulou, and Nikos Deligiannis. "A review on fact extraction and verification." ACM Computing Surveys (CSUR) 55.1 (2021): 1-35.

Bekoulis, Giannis, Christina Papagiannopoulou, and Nikos Deligiannis. "Understanding the Impact of Evidence-Aware Sentence Selection for Fact Checking." Proceedings of the Fourth Workshop on NLP for Internet Freedom: Censorship, Disinformation, and Propaganda. 2021.

Liu, Zhenghao, et al. "Fine-grained Fact Verification with Kernel Graph Attention Network." Proceedings of the 58th Annual Meeting of the Association for Computational Linguistics. 2020.

Author Response

Dear Reviewer 1,

We have tried to improve every comment you have given to us.

We hope that our answers can explain things that are not clear in our paper, and hopefully, this can be a consideration for you to accept our paper.

We attach a word file, which contains our answers. 

Thank you.

Reviewer 2 Report

This manuscript proposes deep bidirectional encoder model called RoBERTaEns for fact verification and especially evaluates homogeneous ensemble model as a bagging model, which suggests that the decision-making of a larger group of people is typically better than that of an individual expert. This manuscript is roughly well-organized, but should be improved as follows:

1) What kind of a bagging model (homogeneous ensemble) do you use? voting, weighted voting, sum and argmax, or soft voting?

2) In bagging ensemble learning, a random sample of data in a training set is selected with/without replacement — meaning that the individual data points can be chosen more than once. In this manuscript, same training set is used for three different base models with different parameters (epoch and batch size). What is the reason?  

3) In Equations and Algorithms,  the variables, RoBERTa_setup, Ma, Na, Ta, etc are unclear.  Are they the number of parameters? or Are they the size of models.  These sound really confusing for readers. 

4) Many sentences are little bits awkward, for example "Sentences have been encoded are matched with ... " on Line 38. 

5) Please give examples for Algorithms and/or Figures in order to easily understand it for readers.  

Author Response

Dear Reviewer,

We have tried to improve every comment you have given to us.

We hope that our answers can explain things that are not clear in our paper, and hopefully, this can be a consideration for you to accept our paper.

We attach a word file, which contains our answers. 

Thank you.

Reviewer 3 Report

In this paper, the authors approach an interesting task for the NLP community, namely Fact Verification.   Although interesting topic, the paper lacks novelty; it does not include any novel idea, technique, or analysis. The solution proposed by the authors reminds me of the articles from the SemEval competition:

  • [1] LI, Junyi; ZHOU, Xiaobing; ZHANG, Zichen. Lee at SemEval-2020 Task 12: A BERT Model Based on the Maximum Self-ensemble Strategy for Identifying Offensive Language. In: Proceedings of the Fourteenth Workshop on Semantic Evaluation. 2020. p. 2067-2072.
  • [2] DAS, Kaushik Amar, et al. KAFK at SemEval-2020 Task 12: Checkpoint ensemble of transformers for hate speech classification. In: Proceedings of the Fourteenth Workshop on Semantic Evaluation. 2020. p. 2023-2029.

As acknowledged by the authors themselves in their paper, the implementation of RoBEFRTaEns is done by combining the results of each RoBERTa with different epoch and batch size settings. The merging process is carried out using the bagging ensemble method.   In some parts, the clarity and editorial quality of the paper weaken. As a consequence, such parts result to be quite difficult to read. Therefore, I would suggest to carefully improve the prose of writing in order to make this paper easier to read.   Moreover, I believe that the Results section should also include a subsection for the Error Analysis and include samples accordingly.   Also, I believe that a section dedicated to parameter exploration and selection will be valuable. A discussion on the parameters selected by the authors would help as well. I would like to see why these specific parameters were chosen along with the authors' rationale for why their settings work best. What experiments led the authors to arrive at their specific combination. To conclude, there is little novelty in this work. All techniques adopted exist already (see papers from SemEval), and no new methods are proposed and tailored for this specific scenario.

Author Response

(The authors gave the same response as above.)

Round 2

Reviewer 1 Report

Changes to the English language here and there before publication

Author Response

Dear Reviewer,

We have revised the manuscript in line with your suggestions.

Please inform us, if there are points that remain we need to improve.

Thank You.

Reviewer 2 Report

This manuscript proposes deep bidirectional encoder model called RoBERTaEns for fact verification and especially evaluates homogeneous ensemble model as a bagging model, which suggests that the decision-making of a larger group of people is typically better than that of an
individual expert. This manuscript is roughly well-organized, but should be improved as follows: 

1) Here is your reply about the bagging model in the previous review:
 "Response 1: The bagging model we use is Sum. As research conducted by Vaswani Ashish, et. al which we mentioned in citation number [39]. They use the Sum method to applying the homogeneuos ensemble to a model."  But The reference is about "Self-Attention", but not bagging. So the previous question on you bagging model is still unclear. Please give the equation of the model to readers. 

2) I'd appreciate it if you could explain the meaning of the Equation (1):  RoBERTa_setup = M_a + N_a < T_a. 
Is the symbol of '+' in Equation (1) is the arithmetic addition? How are they  added? M_a is Model and N_a is a sequence. In addition T_a is a parameter. For readers, you must describe the more details in a concrete way. 

3) The Pseudo-code format on Algorithms should be consistent. In Algorithm 2 though 4, the input and the output are numbered, but not in Algorithm 1. 

Author Response

(The authors gave the same response as above.)

Reviewer 3 Report

In the title, I propose to write "Fact Verification" and not "Fact Veri-fication"

On the page 12, there is many blank line.

Please upload the final version, after corrections, so no red words.

Author Response

(The authors gave the same response as above.)
